# Mental Health Disparities among Pre-Clinical Medical Students at Saint Louis University during the COVID-19 Pandemic

**DOI:** 10.3390/bs14020089

**Published:** 2024-01-26

**Authors:** Won Jong Chwa, Albert C. Chong, Sheryl Lin, Erin H. Su, Chantal Sheridan, Jacob Schreiber, Stephanie K. Zia, Keniesha Thompson

**Affiliations:** 1Department of Medical Education, Saint Louis University School of Medicine, Saint Louis, MO 63104, USA; 2Department of Medical Educaiton, Keck School of Medicine of the University of Southern California, Los Angeles, CA 90033, USA

**Keywords:** anxiety, depression, underrepresented in medicine, Midwest, mental health, medical student, health disparities

## Abstract

The COVID-19 pandemic disproportionately affected racial and ethnic minorities. Medical students were also particularly impacted as they coped with increased stressors due to delayed medical training and a high prevalence of mental health conditions. This study investigates mental health disparities of underrepresented in medicine (URM) students at the Saint Louis University School of Medicine (SLUSOM). An anonymous online survey was distributed to first- and second-year medical students at SLUSOM in February 2021. The survey queried demographic information, lifestyle factors, and pandemic-related and institutional concerns. Mental health was assessed via the Generalized Anxiety Disorder-7 (GAD-7) and the Patient Health Questionnaire-9 (PHQ-9). Statistical tests were run with SPSS, version 27. A convenience sample of 87 students responded to the survey. Students who were categorized as URM were significantly more likely to be at risk of major depressive disorder during the pandemic. Concern about a lack of financial support was significantly greater among students categorized as URM. Concerns regarding a lack of financial support, mental health support, and decreased quality of medical training significantly predicted PHQ-9 scores. Our findings revealed several key factors that may exacerbate mental health disparities among URM students during the pandemic. Providing adequate financial and academic resources for URMs may improve mental health outcomes for similar adverse events in the future.

## 1. Introduction

The COVID-19 pandemic has adversely affected many sectors of society, including the economy, daily life, and human health [1,2]. In particular, the pandemic has disproportionately affected racial and ethnic minorities, exacerbating the health disparities that these groups already face due to higher COVID-19 infection rates and a greater prevalence of pre-existing medical conditions [3,4,5]. Mental health issues, including anxiety and depression, also increased during the pandemic, largely due to the worsening of comorbidities and extensive mass home confinement directives [6]. Thus, the pandemic has widened existing mental health disparities among racial and ethnic minorities [5,7].

Medical students represent another vulnerable demographic due in part to a plethora of environmental stressors such as educational debt, heavy workload, sleep deprivation, information overload, and career planning [8]. As a result, medical students have historically demonstrated higher rates of psychological distress, depression, and anxiety than the general population [8,9,10]. At the height of the pandemic, medical students were subject to delayed clinical rotations and the sudden shift to virtual learning, which resulted in significant change, stress, and uncertainty [11,12]. These changes compounded existing challenges and impacted the mental health of medical students, whose training already required them to adapt to an ever-changing environment while trying to excel in their academic and clinical responsibilities.

Therefore, it is important to investigate the mental health disparities among students who are underrepresented in medicine (URM), as they are at the intersection of two at-risk populations. As defined by the Association of American Medical Colleges, URM refers to racial and ethnic populations that are inadequately represented relative to their proportion in the general population [13]. In our previous study investigating mental health disparities across four academic medical institutions in the west, southeast, and northeast regions of the United States (US), we found disproportionately increased subjective feelings of anxiety and depression between students who identified as URM compared to non-URM during the early periods of the pandemic [14]. However, a limitation of this study was the lack of representation of the Midwest region, an area that has historically experienced greater disparities in healthcare and morbidity compared to other regions [15]. To fill this gap, the present study investigated the impact of the COVID-19 pandemic on medical students at the Saint Louis University School of Medicine (SLUSOM) in Missouri. We aimed to investigate potential mental health disparities among pre-clinical medical students during the pandemic and identify evidence-based solutions for better management of student mental health during global crises. Our research questions were: (1) how do mental health disparities manifest in URM students vs. non-URM students, and (2) what factors may potentially contribute to these disparities. We hypothesized there will be a higher risk of mental health conditions in students identifying as URM due to a multitude of factors, primarily financial burden and a lack of mental health resources.

## 2. Methods

### 2.1. Survey Design and Distribution

The lead author distributed an anonymous online survey via a school listserv to first- and second-year medical students at SLUSOM over a two-and-a-half-week period in February 2021. Participants were informed that the purpose of the survey was to study the effect of COVID-19 on medical student mental health. The survey was electronically administered via Qualtrics. As an incentive, students who completed the survey were invited to participate in a raffle for a chance to win one of three gift cards, each worth USD 10. Contact information was recorded independently from survey responses to ensure confidentiality. Survey protocol was deemed exempt by the Institutional Review Board (IRB) of the University of Southern California (ID# UP-20-01134). The IRB of Saint Louis University also reviewed and agreed with the exemption status and deferred jurisdiction to University of Southern California’s IRB.

The research design was informed by a previously published study with additional questions regarding vaccination status due to the survey at SLUSOM being distributed after vaccines became widely available [14].

The survey included demographic questions about medical school year, household size, distance of home from campus, gender, age, race, and COVID-19 vaccination status. Students who self-identified as African American, American Indian, Alaska Native, Hispanic or Latinx, Native Hawaiian, and other Pacific Islander, including mixes containing any of the aforementioned groups, were coded into the URM category for the purposes of this study. The survey queried various lifestyle factors, including time spent exercising, time spent in clinic, and time spent with friends or family (in person or virtually), all estimated in hours per week. Participants were also asked to rate their levels of concern about institutional factors that may adversely affect mental health, including a lack of academic support, a lack of financial support, a lack of mental health support, delays in medical training, and decreased quality of medical training. Furthermore, participants were asked to rate their level of concern about pandemic-related factors, including a lack of social interaction, worries about contracting and transmitting COVID-19, and new financial difficulty due to the pandemic. Participants were prompted to quantify their level of institutional- and pandemic-related concerns on a scale of 0 to 100. Additionally, participants were given the option to describe other concerns not listed in the survey and score them on the same scale. Finally, pre-pandemic mental health measures were obtained by asking participants whether they had diagnoses of anxiety and/or depression, symptoms, or subjective feelings of these conditions before the pandemic to provide a mental health baseline against anxiety and depression risk during the pandemic.

We administered the Generalized Anxiety Disorder-7 (GAD-7) [16] and the Patient Health Questionnaire-9 (PHQ-9) [17] to assess the risk of clinically significant generalized anxiety disorder (GAD) and major depressive disorder (MDD). The GAD-7, PHQ-9, and questions about mental health history or specific concerns related to the pandemic were included prior to demographic questions to limit potential bias from being prompted about racial/ethnic status. When considering a binary cut-off score for these tests, Spitzer’s study pertaining to the GAD-7 originally denoted a score of 10 as the cutoff for at-risk for anxiety, but a recent meta-analysis demonstrated that any score between 7 and 10 is an acceptable cutoff [16,18]. Similarly, Kroenke initially denoted a score of 10 on the PHQ-9 as the cutoff for at-risk for depression, but a more recent meta-analysis suggested that any score between 8 and 11 is appropriate for diagnosis [17,19]. For this study, we used midpoints in the range of appropriate cutoffs: a score of 9 or higher was categorized as “at-risk” for both GAD and MDD on the GAD-7 and PHQ-9. Stratified analyses were performed, with cutoffs for severe anxiety and depression with GAD-7 and PHQ-9 at 15 and 20, respectively [16,17].

### 2.2. Statistical Analysis

We performed statistical analyses with the IBM Statistical Package for the Social Sciences (SPSS), Version 27 (IBM Corp, Armonk, NY, USA). We applied chi-squared tests to investigate categorical differences between URM status and GAD and MDD risk during the pandemic era, as well as to compare pre-existing diagnoses of depression or anxiety between groups before the pandemic. Chi-squared tests were also used to test whether GAD or MDD risk was associated with a spectrum of demographic characteristics. We employed Mann–Whitney U tests to analyze differences in levels of pandemic-related and institutional concerns between students in the URM group and non-URM group. For these tests, participants who did not denote a racial or ethnic category were excluded from the analysis. Finally, we performed multiple linear regression across all respondents to predict GAD and MDD scores based on institutional- and pandemic-related concerns.

## 3. Results

### 3.1. Demographics

Out of 366 pre-clinical medical students at SLUSOM, a convenience sample of 87 students (24%) responded to the survey. Of these, 32 (36.8%) identified as male, 54 (62.1%) identified as female, and 1 (1.1%) preferred not to identify their gender. Regarding racial and ethnic demographics, 73 (83.9%) students identified as a race or ethnicity categorized as non-URM, 11 (12.6%) students as URM, and 3 (3.4%) preferred not to identify their race or ethnicity. With regards to school year, 55 (63.2%) were first-year medical students, while 32 (37.8%) were second-year students (Table 1).

### 3.2. URM vs. Non-URM Demographic Characteristics and Mental Health Outcomes

Chi-squared analyses revealed a statistically significant difference between the proportion of students categorized as URM versus those not categorized as URM at risk of depression on the PHQ-9 (X^2^(2) = 4.184, df = 2, *p* = 0.041, Figure 1). During the COVID-19 era, the proportion of respondents in the URM group at risk of MDD (54.5%, n = 11, Table 2) was significantly higher than the proportion of students in the non-URM group (24.7%, n = 73, Table 2). An odds ratio was calculated to interpret this difference, which illustrated that those categorized as URM were 3.67 times more likely to be at risk for depression during the COVID-19 pandemic than those categorized as non-URM. The proportion of respondents in the URM category at-risk of GAD (45.5%, n = 11, Table 2) was not significantly different from the proportion of respondents in the non-URM group (27.4%, n = 73, Table 2) (X^2^(2) = 1.491, df = 2, *p* = 0.222, Figure 1).

Among those who were at risk of MDD, chi-squared analyses revealed no statistically significant difference between the proportion of respondents identifying as URM categorized as having severe MDD (14.3%, n = 7, Appendix A) versus respondents identifying as non-URM (10.3%, n = 29, Appendix A) (X^2^(2) = 0.0887, df = 2, *p* = 0.766, Appendix A). Among those who were at risk of GAD, the proportion of respondents identifying as URM categorized as having severe GAD (28.6%, n = 7, Appendix A) was not significantly different from respondents identifying as non-URM (17.2%, n = 29, Appendix A) (X^2^(2) = 0.4621, df = 2, *p* = 0.497, Appendix A).

Chi-squared analyses of the URM category by pre-pandemic health outcomes showed no statistically significant association between students in the URM category and those in the non-URM category by their subjective feelings of anxiety (X^2^(2) = 2.615, df = 2, *p* = 0.106) or sadness (X^2^(2) = 0.035, df = 2, *p* = 0.852) before the pandemic. Similarly, there was no statistically significant association between the proportion of students in the URM category and those who were not categorized as URM who had a pre-existing diagnosis of anxiety (X^2^(2) = 0.001, df = 2, *p* = 0.976) or depression (X^2^(2) = 0.814, df = 2, *p* = 0.367).

There were no statistically significant differences between students in the URM and non-URM categories by vaccination status (X^2^(2) = 2.349, df = 2, *p* = 0.359), history of contracting COVID-19 (X^2^(2) = 2.588, df = 2, *p* = 0.651), number of people in household (X^2^(2) = 5.776, df = 6, *p* = 0.449), virtual interaction hours (X^2^(2) = 3.751, df = 8, *p* = 0.778), or in-person interaction hours (X^2^(2) = 4.184, df = 8, *p* = 0.864).

### 3.3. Demographic Predictors of GAD/MDD Risk

Chi-squared tests were performed to determine whether other categorical demographic variables besides URM status were associated with GAD or MDD risk. These variables included medical school year, household size, distance of home from campus, gender, age, and COVID-19 vaccination status. However, none of these factors significantly predicted risk for either GAD or MDD.

### 3.4. Concerns among URM vs. Non-URM Students

Concern about lack of financial support (U = 232.500, *p* = 0.024) was significantly greater among students categorized in the URM group compared to those in the non-URM group during the pandemic era. Concerns about decreased quality of training (U = 262.500, *p* = 0.065), a lack of academic support (U = 281.000, *p* = 0.109), a lack of mental health support (U = 337.500, *p* = 0.394), and delays in medical training (U = 296.500, *p* = 0.162) were not statistically significant between students in the URM group and non-URM group.

Differences in pandemic-related concerns, including lack of social interaction (U = 349.500, *p* = 0.490), worries about contracting (U = 360.500, *p* = 0.586) and transmitting COVID-19 (U = 345.500, *p* = 0.457), and new financial difficulties due to the pandemic (U = 268.000, *p* = 0.073) were not statistically significant between students categorized in the URM group and the non-URM group.

### 3.5. Associations between Concerns and PHQ-9/GAD-7 Scores

Multiple linear regression was performed to identify institutional predictors of PHQ-9 and GAD-7 scores. Full statistical details of the multiple regression analyses are listed in Appendix A. The model significantly predicted PHQ-9 scores, F(5,81) = 5.965, *p* < 0.001, explaining 26.9% of the variance in the outcome. Concerns about financial support (Std β = 0.251, *p* = 0.021) and concerns about the quality of medical training (Std β = 0.314, *p* = 0.035) significantly predicted PHQ-9 scores. Both significant relationships were positive. Concerns about academic support (*p* = 0.546), mental health support (*p* = 0.066), and delayed medical training (*p* = 0.575) did not significantly predict PHQ-9 scores. Furthermore, the model significantly predicted GAD-7 scores, F(5,81) = 4.939, *p* < 0.001, explaining 23.4% of the variance in the outcome. Concerns about mental health support (Std β = 0.347, *p* = 0.030) significantly predicted GAD-7 scores in a positive relationship. Concerns about academic support (*p* = 0.520), financial support (*p* = 0.860), delayed medical training (*p* = 0.111), and quality of medical training (*p* = 0.071) did not significantly predict GAD-7 scores (Appendix A).

Multiple linear regression was also performed to identify pandemic-related predictors of PHQ-9 and GAD-7 scores. The model significantly predicted PHQ-9 scores, F(4,82) = 6.779, *p* < 0.001, explaining 24.8% of the variance in the outcome. Concerns about the lack of social interaction (Std β = 0.255, *p* = 0.014) and new financial difficulties (Std β = 0.387, *p* < 0.001) significantly predicted PHQ-9 scores. Both significant relationships were positive. Concerns about contracting COVID-19 (*p* = 0.422) and transmitting COVID-19 (*p* = 0.463) did not significantly predict PHQ-9 scores. The model also significantly predicted GAD-7 scores, F(4,82) = 2.808, *p* = 0.031, explaining 12.0% of the variance in the outcome. Concerns about new financial difficulties (Std β = 0.226, *p* = 0.039) significantly predicted GAD-7 scores in a positive relationship. Concerns of a lack of social interaction (*p* = 0.200), contracting COVID-19 (*p* = 0.499), and transmitting COVID-19 (*p* = 0.765) did not significantly predict GAD-7 scores (Appendix A).

## 4. Discussion

Our results suggest increased mental health disparities during the COVID-19 pandemic among students who were categorized as URM in this sample. Primarily, those in the URM group were significantly more likely to be at risk of MDD than students in the non-URM group. There was also an observed but not statistically significant difference in GAD risk between the two groups. In our previous study covering areas besides the Midwest, URM status was significantly associated with GAD but not MDD [14]. Considering the direction of associations in both studies, URMs may demonstrate higher risk in both GAD and MDD. Other studies have also shown similar deteriorations in mental health among medical students before and during the pandemic [20,21]. These disparities may be attributed to perceived differences in learning experiences and quality of life [9,10,22,23,24]. Such differences may be perpetuated due to the trend of students who identify with minoritized groups experiencing less access to mental health services and a lower likelihood of utilizing mental health resources [25]. Yet, these differences were not present in students’ self-reported subjective symptoms and pre-existing diagnoses of depression or anxiety before the pandemic, suggesting that the pandemic may have exacerbated mental health disparities between URM and non-URM students. Therefore, institutions should monitor and target interventions to address mental health disparities for students who identify as URM during times of great adversity.

Consistent with our previous study, the data suggest that students who identified with groups categorized as URM were significantly more concerned than their non-URM peers about a lack of financial support [14]. Financial concern also significantly predicted PHQ-9 scores. Given that socioeconomic disparities between students who identify with groups of a minoritized background and those in the majority are well-documented [26], such concerns likely play a role in the mental health of URMs. Though not significantly different between groups, those who identified as URM were more concerned about the perceived decreased quality of medical training. This concern also significantly predicted PHQ-9 scores, which could be attributed to higher attrition rates, less exposure to the healthcare field, and lower accessibility to academic support compared to non-URMs [27]. Furthermore, “stereotype threat” may be another contributing factor, in that personal experiences of discrimination and pressure to dispel negative stereotypes add to the academic and psychological burden of URMs [28]. Therefore, the perception of decreased quality of medical training may put an even greater mental burden on URMs facing these challenges.

Concerns regarding mental health support were not significantly different between those who were categorized as URMs and non-URMs, which differs from our previous study [14]. This may be partially attributed to variations in measures implemented by medical schools to address pandemic-related concerns, such as virtual learning and well-being lectures, which, in some settings, may have varying impacts on student mental health. [29,30,31]. Notably, the survey from SLUSOM was administered two months after the other institutions involved in the previous study, which allowed more time for the institution to adapt and implement resources for medical students. A key example is seen within the last several months prior to survey distribution: the expansion of SLUSOM’s Office of Diversity, Equity, and Inclusion, a collaborative department to promote and cultivate diversity and inclusion, particularly for the underrepresented [32].

Given these findings, placing a strong emphasis on providing adequate financial and academic resources for students in URM groups, such as additional scholarship opportunities, mentorship, and extracurricular support networks, may improve mental health outcomes [28] For example, a qualitative study revealed that medical students in London found cash stipends and housing credits reduced stress, and many liked the idea of institution-provided access to peer or professional financial advisors [33]. Medical schools may also consider collaborating with the Liaison Committee on Medical Education (LCME) and the Association of American Medical Colleges (AAMC) for guidance to promptly adapt curriculum based on the needs of students and faculty [34], helping to ensure that medical schools maintain a high quality of education even during extenuating circumstances.

This study has several limitations. First, the use of a convenience sample may have resulted in an increased number of responses from students who were either interested in student wellness or experienced a greater impact on their mental health due to the pandemic. Second, the difference in sample size between students who were categorized as non-URMs (n = 73) and URMs (n = 11) may limit statistical power. However, it is worth noting that the proportion of students categorized as URM in this study is similar to the approximately 14% of medical students nationally who identified as URM in 2020–2021, and 10.3% within SLUSOM [35]. Third, a greater proportion of women responded to the survey relative to men, though this outcome may reflect increased prevalence of mental health concerns among women in the general population [21,36]. Fourth, the small sample size also resulted in smaller effect sizes for many of our findings, and, due to low categorical counts, we were not able to further stratify GAD or MDD based on severity using the screening tools. Finally, the cross-sectional nature of this study limits inferences about temporal variables such as vaccination status and baseline mental health. We attempted to control these measures through investigation of vaccination status and previous diagnoses of anxiety and depression, in which no significant relationships between URM and non-URM variables were observed.

## 5. Conclusions

Together, our findings reveal several key factors that may exacerbate mental health disparities among students identifying as URM during the pandemic. This study contributes novel data regarding mental health disparities for students in URM groups at a midwestern US medical school during the height of the COVID-19 pandemic and offers potential strategies to reduce disparities and improve student wellbeing should we face similar adverse events in the future.

## Figures and Tables

**Figure 1 behavsci-14-00089-f001:**
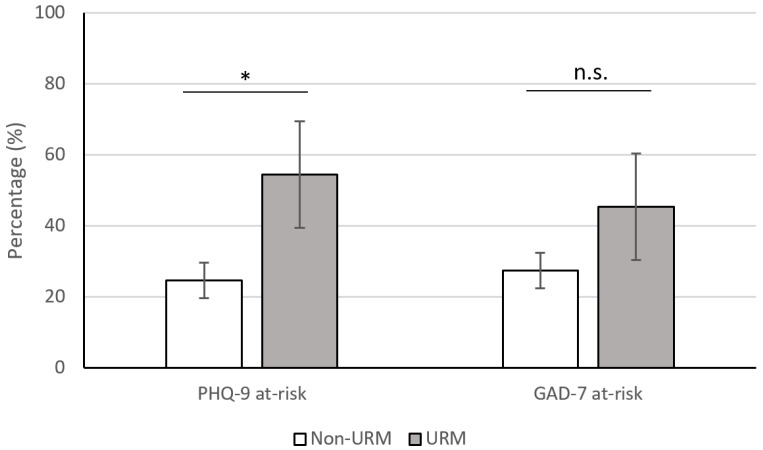
Left: Percentage of survey participants who were not at risk or at risk of depression based on PHQ-9 scores, with a threshold of 9. Right: Percentage of survey participants who were not at risk or at risk of anxiety based on GAD-7 scores, with a threshold of 9. Statistical significance was calculated via chi-squared analyses. Error bars indicate standard error. PHQ-9: Patient Health Questionnaire-9. GAD-7: Generalized Anxiety Disorder-7. URM = underrepresented in medicine. * = Statistically significant, *p* = 0.04; n.s. = not significant, *p* = 0.22.

**Table 1 behavsci-14-00089-t001:** Demographic characteristics of survey sample, stratified by non-URM, URM, and other respondents. URM = underrepresented in medicine.

	URM (*n* = 11)	Non URM (*n* = 73)	Prefer Not to Answer (*n* = 3)
Age			
20–22	1 (9.1%)	14 (19.2%)	0 (0%)
23–25	8 (72.7%)	52 (71.2%)	3 (100%)
26–28	2 (18.2%)	5 (6.8%)	0 (0%)
29+	0 (0%)	2 (2.7%)	0 (0%)
Gender			
Male	4 (36.4%)	27 (37%)	1 (33.3%)
Female	7 (63.6%)	46 (63%)	1 (33.3%)
Transgender	0 (0%)	0 (0%)	0 (0%)
Prefer Not to Answer	0 (0%)	0 (0%)	1 (33.3%)
Race			
Asian		23 (31.5%)	
White		48 (65.8%)	
Mixed (non-URM)		2 (2.7%)	
Black or African American	4 (36.4%)		
Hispanic/Latinx/Spanish origin	2 (18.2%)		
Mixed (URM)	5 (44.5%)		
Prefer not to Answer			3 (100%)
Medical School Level			
M1	8 (73%)	47 (64.4%)	0 (0%)
M2	3 (27.3%)	26 (35.6%)	3 (100%)
Contracted COVID-19			
Yes	2 (18.2%)	7 (9.6%)	0 (0%)
No	7 (63.6%)	56 (76.7%)	2 (66.7%)
Unsure	2 (18.2%)	10 (13.7%)	1 (33.3%)
Vaccination Status			
Yes (2 doses)	2 (18.2%)	23 (31.5%)	1 (33.3%)
Yes (1 dose)	0 (0%)	4 (5.5%)	1 (33.3%)
No	9 (82.8%)	46 (63%)	1 (33.3%)
Distance from Campus			
0–5 Miles	10 (91.9%)	63 (86.3%)	3 (100%)
6–10 Miles	1 (9.1%)	6 (8.2%)	0 (0%)
11–15 Miles	0 (0%)	3 (4.1%)	0 (0%)
16–20 Miles	0 (0%)	1 (1.4%)	0 (0%)
Time in Clinic (per week)			
0–2 h	7 (63.6%)	51 (69.9%)	1 (33.3%)
3–5 h	3 (27.3%)	19 (26%)	2 (66.7%)
6–8 h	1 (9.1%)	1 (1.4%)	0 (0%)
9–11 h	0 (0%)	1 (1.4%)	0 (0%)
12+ h	0 (0%)	1 (1.4%)	0 (0%)
Other People in Household			
0	7 (63.6%)	23 (31.5%)	2 (66.7%)
1	2 (18.2%)	30 (41.1%)	0 (0%)
2	1 (9.1%)	13 (17.8%)	1 (33.3%)
3	1 (9.1%)	7 (9.6%)	0 (0%)
Exercise (per week)			
0–2 h	7 (63.6%)	24 (32.9%)	2 (66.7%)
3–5 h	3 (27.3%)	35 (47.9%)	1 (33.3%)
6–8 h	1 (9.1%)	12 (16.4%)	0 (0%)
9–11 h	0 (0%)	2 (2.7%)	0 (0%)
Virtual Interaction			
0–2 h	4 (45.5%)	25 (34.2%)	2 (66.7%)
3–5 h	4 (45.5%)	29 (39.7%)	0 (0%)
6–8 h	3 (27.3%)	12 (16.4%)	1 (33.3%)
9–11 h	0 (0%)	6 (8.2%)	0 (0%)
12+ h	0 (0%)	1 (1.4%)	0 (0%)
In-Person Interaction			
0–2 h	6 (54.5%)	26 (35.6%)	2 (66.7%)
3–5 h	2 (18.2%)	26 (35.6%)	0 (0%)
6–8 h	1 (9.1%)	11 (15.1%)	1 (33.3%)
9–11 h	1 (9.1%)	5 (6.8%)	0 (0%)
12+ h	1 (9.1%)	5 (6.8%)	0 (0%)

**Table 2 behavsci-14-00089-t002:** Mental health characteristics of survey sample, stratified by non-URM, URM, and other respondents. A cut-off score of 9 or greater on the PHQ-9 or GAD-7 was denoted as “at-risk” for depression or anxiety, respectively. URM = underrepresented in medicine. Dx = diagnosis. PHQ-9: Patient Health Questionnaire-9. GAD-7: Generalized Anxiety Disorder-7.

	URM (*n* = 11)	Non-URM (*n* = 73)	Prefer Not to Answer (*n* = 3)
Subjective anxiety before 2019			
Yes	3 (27%)	39 (53.4%)	1 (33.3%)
No	8 (73%)	34 (46.6%)	2 (66.7%)
Anxiety Dx before 2019			
Yes	2 (18%)	13 (17.8%)	1 (33.3%)
No	9 (82%)	60 (82.2%)	2 (66.7%)
Subjective depression before 2019			
Yes	3 (27%)	18 (24.7%)	1 (33.3%)
No	8 (73%)	55 (75.3%)	2 (66.7%)
Depression Dx before 2019			
Yes	1 (9%)	15 (20.5%)	1 (33.3%)
No	10 (91%)	58 (79.5%)	2 (66.7%)
PHQ-9 Risk			
At Risk	6 (54.5%)	18 (24.7%)	0 (0%)
Not at Risk	5 (45.5%)	55 (75.3%)	3 (100%)
GAD-7 Risk			
At Risk	5 (45.5%)	20 (27.4%)	0 (0%)
Not at Risk	6 (54.5%)	53 (72.6%)	3 (100%)

## Data Availability

The data presented in this study are available on request from the corresponding author. The data are not publicly available due to ethical and privacy considerations.

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
