# Peer review of "Mental Health Disparities among Pre-Clinical Medical Students at Saint Louis University during the COVID-19 Pandemic"

_behavsci, 2024, doi:10.3390/bs14020089_

Round 1

Reviewer 1 Report

Comments and Suggestions for Authors

This is a thorough analysis of GAD and MDD outcomes in a convenience sample from a single midwestern medical school. The authors are interested in whether URM students reported worse outcomes on these scales during the Covid pandemic. If so, they are interested in what factors beyond URM status correlate with those outcomes and what sources of concern students report.

Since the sample is small, particularly the subsample of interest (N=11), there is little additional analysis that is likely to yield useful insights. However, I’d suggest the authors explore severe MDD or GAD outcomes as separate from “any” depression or anxiety. This simply means defining a higher cutoff for these diagnoses and performing the analysis in Figure 1 and Table 2 using this additional definition. This would supplement, not replace, the Figure 1/Table 2 analysis.

The authors may then want to explore whether there are predictors (correlates) of high cut-off diagnoses (risk) or whether associated concerns differ when a higher threshold is used.

Reviewer 2 Report

Comments and Suggestions for Authors

Thank you for the opportunity to review this manuscript. The content will be of great interest to educators and medical staff involved in teaching. In order to improve the script the following is suggested:

Under the methodology section there needs to be some discussion on research design and ethical considerations that is who granted ethical permission for this study and was a gatekeeper used for example to distribute the questionnaire used. What informed the design of the questionnaire? 
